# Anti-Metalloproteases: Production and Characterization of Polyclonal IgG Anti-F2 Fraction Antibodies Purified from the Venom of the Snake *Bitis arietans*

**DOI:** 10.3390/toxins15040264

**Published:** 2023-04-01

**Authors:** Kemily Stephanie de Godoi, Felipe Raimondi Guidolin, Fernanda Calheta Vieira Portaro, Patrick Jack Spencer, Wilmar Dias da Silva

**Affiliations:** 1Immunochemistry Laboratory, Butantan Institute, São Paulo 05503-900, Brazil; 2Laboratory of Structure and Function of Biomolecules, Butantan Institute, São Paulo 05503-900, Brazil; 3Biotechnology Center, Nuclear and Energy Research Institute, São Paulo 05508-000, Brazil

**Keywords:** antivenoms, *Bitis arietans*, polyclonal antibodies, metalloproteases

## Abstract

*Bitis arietans* is a medically important snake found in Sub-Saharan Africa. The envenomation is characterized by local and systemic effects, and the lack of antivenoms aggravates the treatment. This study aimed to identify venom toxins and develop antitoxins. The F2 fraction obtained from *Bitis arietans* venom (BaV) demonstrated the presence of several proteins in its composition, including metalloproteases. Titration assays carried out together with the immunization of mice demonstrated the development of anti-F2 fraction antibodies by the animals. The determination of the affinity of antibodies against different *Bitis* venoms was evaluated, revealing that only BaV had peptides recognized by anti-F2 fraction antibodies. In vivo analyses demonstrated the hemorrhagic capacity of the venom and the effectiveness of the antibodies in inhibiting up to 80% of the hemorrhage and 0% of the lethality caused by BaV. Together, the data indicate: (1) the prevalence of proteins that influence hemostasis and envenomation; (2) the effectiveness of antibodies in inhibiting specific activities of BaV; and (3) isolation and characterization of toxins can become crucial steps in the development of new alternative treatments. Thus, the results obtained help in understanding the envenoming mechanism and may be useful for the study of new complementary therapies.

## 1. Introduction

Snakebite is considered a serious public health problem, affecting individuals all over the world. Snakebites excessively affect the areas in tropical and subtropical countries located in Asia, Africa, Oceania, and Latin America, causing high rates of morbidity and mortality [1,2]. The regions with the highest occurrence of accidents with venomous snakes are Sub-Saharan Africa, South and Southeast Asia, Papua New Guinea (Oceania), and Latin America. In the African continent, this rate varies from 3500 to 32,000 deaths per year [3].

Most snake bites in Sub-Saharan Africa are caused by snakes of the genus Bitis, which belong to the Viperidae family and are widely distributed on the continent [4]. The genus *Bitis* includes 17 species of snakes found in Africa and Arabia [5,6]. Among these snakes, the Bitis genus includes four species, which, due to the number and severity of envenomation incidents, are important: *B. arietans*; *B. gabonica*; *B. rhinoceros*; and *B. nasicornis* [7,8]. 

The snake *B. arietans* is one of the most medically important snakes present on the African continent and can also be found in Sub-Saharan Africa and in savannas and pastures in Morocco and Arabia, with the exception of the Sahara region and tropical forests. This species inhabits a variety of habitats, especially in regions with large human and animal populations [4,7].

*B. arietans* envenomation causes local effects, such as intense pain, blister formation, swelling, ecchymosis, hemorrhage at the bite site, necrosis, and dilation of lymph nodes in the affected region. Systemic effects include fever, neutrophilic leukocytosis, thrombocytopenia, anemia, hypotension, and bradycardia. All of these can cause death [4,8,9,10,11,12,13,14,15,16]. 

*Bitis arietans* venom (BaV) is composed of proteins, peptides, metal cations, carbohydrates, nucleosides, biogenic amines, free amino acids, and lipids [17]. Proteome and transcriptome analyses revealed that the venom of *B. arietans* and other *Bitis* spp. are made up of proteins from the following families: metalloproteases (SVMPs) (38.5%); serine proteases (SVSPs) (19.5%); phospholipases (PLA2) (4.3%); disintegrins (17.8%); C-type lectins (CTL) 13.2%); Kunitz type (4.1%); and cystatins (1.7%) [18,19,20,21]. The abundant presence of bradykinin-enhancing peptides (BPPs) in the BaV was experimentally demonstrated, and they were also found in other snakes of the genus Bitis [22]. Among the most common components of BaV are proteases, especially SVMPs and SVSPs, which represent 58% of the venom composition [18,20,21].

As *B. arietans* presents not only a wide range based on geographic location [23] but also differences in venom composition, there are enormous difficulties in promoting treatment for snake envenomation. In West Africa, accident victims have hemorrhage and petechiae attributed to thrombocytopenia without the occurrence of coagulopathies [12]. On the other hand, accident victims in East and Southern Africa presented disturbances in the blood clotting process attributed to coagulopathy [24]. On the recommendation of the World Health Organization (WHO), for envenomation by *B. arietans*, a specific polyvalent antivenom (produced from the venoms of *B. arietans*, *B. gabonica*, and *B. rhinoceros*) should be administered intravenously as soon as the symptoms appear [4]. Even with a high prevalence of accidents involving *B. arietans*, the biochemical mechanisms involved are little known. The precariousness of investment in specific treatments for accidents with snakes on the African continent justifies the development of studies on the components and mechanism of action of the venom, with the aim of understanding the biological processes involved and establishing complementary therapies quicker.

Due to the importance of these molecules, our group has been working on the purification and characterization of BaV. Thus, initially, the F2 fraction containing different types of proteins, such as SVMPs, snaclecs, and hyaluronidases, was purified from BaV and characterized in vitro and in vivo. Snaclec and hyaluronidase directly participate in the toxic actions caused by SVMPs [25,26,27].

Based on the importance of the hemorrhagic process caused by viperid envenomation and being frequently associated with metalloproteinases in the vasculature [28,29], in vivo hemorrhage induced by BaV was studied subcutaneously in mice. In general, this study demonstrated that BaV contains toxins that trigger the hemorrhagic process, in addition to toxins that help in the spreading factor of the toxic activity of one of the main proteases of venom, the metalloproteinases.

## 2. Results

### 2.1. The Affinity Column Fractionation and Electrophoretic Profile of the BaV and F2 Fraction

The *Bitis arietans* venom (BaV) was fractionated by an affinity chromatography step on an immobilized zinc column. The peak-containing material not retained by the column was identified as F1, and the peak containing the material of interest was identified as F2 (Figure 1A). After dialysis, the F2 fraction was subjected to protein quantification by the bicinchoninic acid (BCA) method [30], using the commercial Pierce BCA Protein Assay kit (Rockford, IL, USA), which resulted in a concentration of 1418 µg/mL. For data intercalation, a standard curve was used with concentrations of 0–2000 µg/mL of bovine serum albumin (BSA) (Sigma–Aldrich, St. Louis, MO, USA) diluted in PBS pH 7.2. At 540 nm; the reading was taken with an ELX 800 plate spectrophotometer (Biotek Instruments, Winooski, VT, USA) (Table 1). 

The electrophoretic analysis of the BaV profile and the F2 fraction revealed the presence of proteins with molecular mass ranging from 135 to 17 kDa and 50 and 17 kDa, respectively. Both samples were revealed by silver impregnation. (Figure 1B). 

### 2.2. Proteolytic Activity of the F2 Fraction

The proteolytic activity of the F2 fraction was evaluated through cleavage by FRET substrates—Abz-FRSSF-EDDnp and Abz-RPPGFRSPFR-QEDDnp. Table 1 demonstrates that the F2 fraction, as well as the BaV used for the positive control, has enzymatic activity against the substrates tested. Partial validation of the presence of SVMPs in the F2 fraction can be confirmed by cleavage with the substrate Abz-RPPGFRSPFR-QEDDnp. According to the table, the average reaction rate of the enzymatic activity demonstrates the reaction factor of the BaV and the F2 fraction against specific substrates. Was observed lower activity of serine proteases (SVSPs) in the F2 fraction, even in the presence of a specific substrate, while the activity of metalloproteases (SVMPs) in the F2 fraction together with the specific substrate revealed a high prevalence of these proteases in the sample.

### 2.3. Selective Inhibition of the Proteolytic Activity of BaV and the F2 Fraction with Specific Inhibitors

The selective inhibition of SVMPs (EDTA) and SVSPs (PMSF) under the Abz-RPPGFRSPFR-QEDDnp substrate evaluated in a 15 min kinetics revealed the presence of these proteases in BaV and in F2 fraction. F2 fraction had the highest concentration of SVMPs (Table 2). These results, together with the enzyme activity test, confirm the presence of SVMPs in the F2 fraction. The average reaction velocity calculated from the plateau of each curve of the samples demonstrates the decrease in the speed of action of proteases against specific inhibitors, resulting in a higher inhibition index and confirming the presence of SVMPs in the samples.

### 2.4. Identification of F2 Fraction Proteins by Mass Spectrometry

F2 fraction was submitted to digestion with trypsin in solution and later identified by mass spectrometry (LC-MS/MS). The data obtained were submitted to bioinformatic analysis using BLASTP 2.13.0 [31] and UniProtKB/SwissProt, and submitted for a search in the “Serpentes” database (taxid: 8570). A total of 57 peptides were found in the F2 fraction (Table 3). The most abundant proteins in the fraction were SVMPs (35%), hyaluronidase (31%), and snaclecs/C-type lectins (21%). Serine proteases (SVSPs) (5%), PLI alpha (PLA2 inhibitor) (2%), cysteine-rich venom protein (2%), AdTx1 (muscarinic toxin 3) (2%), and VEGF-Fs (endothelial growth factor toxin vascular venom), Barietin (2%), was found in relatively low abundance.

### 2.5. Immunoreactivity Analysis of Anti-F2 Fraction Antibodies

The titration of the plasmas obtained by the immunization process was analyzed using ELISA, as described in Section 5.7. The plates sensitized with 1 μg/well of antigen, and the plasma dilutions ranged from 1:100 to 1:12,800. To elucidate the results of each immunization cycle, the average of the 10 mice used was calculated. As shown in Table 4, all mice developed anti-F2 fraction antibodies, an evolution that demonstrates the maximum titers obtained in U/mL in each immunization step.

The yield of antibodies at each stage of immunization was determined by the ELISA method in 96-well plates sensitized with one µg of antigen per well. The sera were serially diluted (1:20 to 1:10,240) in PBS/BSA 0.1%. Detection with peroxidase-conjugated “anti-mouse” antibodies was performed at a dilution of 1:7500. The reading was taken with ELX 800 plate spectrophotometer (Biotek Instruments, Vermont, USA) at a wavelength of 490 nm. The yield was presented as units per milliliter (U/mL). The assay was performed in duplicate.

### 2.6. Cross-Recognition by ELISA 

Cross-recognition titration was performed using ELISA against *B. gabonica*, *B. nasicornis*, and *B. rhinoceros* venoms, using the anti-F2 fraction antibodies with dilutions ranging from 1:500 to 1:256,000. It was not possible to obtain enough basal numbers to establish a yield, demonstrating that the anti-F2 fraction antibodies do not recognize the peptides present in the venom of *B. gabonica*, *B. nasicornis*, and *B. rhinoceros*. On the other hand, it was possible to obtain a yield of 37,000 EU/mL from the anti-F2 fraction antibodies against BaV (Figure 2). The assay was performed in duplicate.

### 2.7. Cross-Recognition by Western Blot

The venoms of *B. arietans*, *B. gabonica*, *B. nasicornis* and *B. rhinoceros*, in addition to the F2 fraction, were resolved as described in Section 5.10. The venoms (5 µg) were treated with non-reducing SDS-PAGE resolved in 5% for upper gel and 12% for lower gel. After electrophoresis, samples were either stained with silver (Figure 3A) or transferred to a nitrocellulose membrane for western blot (Figure 3B), and recognition of the bands by anti-F2 fraction antibodies (1:200) was confirmed. A positive response was observed against bands presents in the F2 fraction equivalent to 50 and 26 kDa. Bands located between 100 and 20 kDa were observed in BaV. Poor recognition was observed for ≈52 kDa bands in *B. gabonica*, *B. nasicornis* and *B. rhinoceros* venoms, as well as poorly recognized 34 kDa bands. Subsequently, the bands equivalent to 50 and 26 kDa of F2 fraction (F2.1, F2.2), 52 kDa of *B. gabonica* (Bg), 52 kDa of *B. nasicornis* (Bn), and 52 kDa of *B. rhinoceros* (Br) were selected for analysis by mass spectrometry.

### 2.8. Identification of the Peptides Corresponding to the Bands of the Venoms of Snakes by Mass Spectrometry

Selected bands from the polyacrylamide gel (Bg, Bn, Br, F2.1 and F2.2) obtained in Section 2.7 was submitted to in gel digestion and identified by mass spectrometry. The data obtained were submitted to bioinformatic analysis using BLASTP 2.13.0 [31], UniProtKB/SwissProt, and PEAKS DB, and submitted to a search in the “Serpentes” database (taxid: 8570). The Appendix A lists the most abundant proteins found in each analyzed sample. The low presence of peptides, such as SVMPs, disintegrin, and snaclecs, explains the poor recognition of bands from *B. gabonica*, *B. nasicornis*, and *B. rhinoceros* by anti-F2 fraction antibodies. In the F2 fraction, it was possible to identify peptides from the SVMPs and Snacles families, in addition to peptides that match the composition of the venom.

### 2.9. Hyperimmune Plasma Affinity Determination

Affinity determination was performed by ELISA with the inclusion of an elution step with KSCN at concentrations ranging from 0 M to 5 M. The anti-F2 fraction antibodies were used at a fixed dilution of 1:1000. The affinity curves of the F2 fraction and the BaV are similar and demonstrate that the preformed complexes of antibodies + antigens resist in the presence of concentrations of KSCN at 3 M, interrupted thereafter (Figure 4A,B). The determination of the affinity index, calculated from the 3 M concentration of KSCN, showed a similar percentage of bound antibodies between the groups, with an affinity index of 31% and 28% against the F2 fraction and the crude venom, respectively, proving that there was no statistically significant difference between the BaV and the F2 fraction (Figure 4C).

### 2.10. Inhibition of the Proteolytic Activity of BaV and the F2 Fraction with Anti-F2 Fraction Antibodies

The selective inhibition of SVMPs present in the BaV and F2 fractions with anti-F2 fraction antibodies used as inhibitors and evaluated in a 15 min kinetics revealed the antibodies inhibitory capacity against these proteases (Table 5). The average speed of reaction calculated from the plateau of each curve of the samples shows a decrease in the speed of action of proteases against the antibodies used as inhibitors, resulting in a higher index of inhibition.

### 2.11. Hemorrhagic Activity Induced by BaV

The induction of hemorrhage by the BaV was tested by intradermal injection with different venom concentrations (10, 20, 30, or 40 µg/animal) or with PBS pH 7.2 as a negative control. After 3 h, the animals were sacrificed intraperitoneally with an overdose of the anesthetics Xylazine (30 mg/kg) and Ketamine (300 mg/kg). The skin was dissected, and the hemorrhagic areas were exposed on the inner surface of the skin. Figure 5A indicates that, initially, at 10 µg, hemorrhage has already developed to a large extent. When compared with the injections of 20, 30, and 40 µg/animal, it is possible to show a wide difference in the hemorrhagic area (Figure 5B–D). Animals inoculated only with PBS pH 7.2 used as a negative control showed no interference in the assay (Figure 5E). These results were compatible when the corresponding areas were measured and expressed in square millimeters of hemorrhagic area. A1: 231 mm^2^, A2: 263 mm^2^; B1: 287 mm^2^, B2: 579 mm^2^; C1: 319 mm^2^, C2: 340 mm^2^; D1: 650 mm^2^, D2: 396 mm^2^ (Table 6). 

### 2.12. Inhibition of Hemorrhagic Activity by Anti-F2 Fraction Antibodies

The serum neutralization of BaV hemorrhagic activity was determined by injecting the minimum hemorrhagic dose (MHD) with 10 µg/animal of BaV and incubating it with different concentrations of the anti-F2 fraction antibody (1:5, 1:10, and 1:20). After 3 h, the animals were sacrificed intraperitoneally with an overdose of the anesthetics Xylazine (30 mg/kg) and Ketamine (300 mg/kg). The skin was dissected and the hemorrhagic areas exposed on the inner surface of the skin. As shown in Figure 6, it is possible to observe a reduction in the hemorrhagic halo where anti-F2 fraction antibodies were administered at 1:5 (Figure 6A) and 1:10 (Figure 6B) concentrations. Figure 6C shows no difference in the hemorrhagic halo when compared to the control group (Figure 6D). These results were compatible when measuring the corresponding areas, expressed in square millimeters of halo hemorrhagic area and medium percentage of reduced area (Table 7).

### 2.13. Inhibition of Lethality by Anti-F2 Fraction Antibodies

In vivo protection (ED50) was determined by injecting 2 LD_50_ (100 µg/animal) of BaV diluted in PBS pH 7.2, incubated with different concentrations of the anti-F2 fraction antibody (1:10, 1:20, and 1:40), into four groups of animals (n = 16). It was experimentally demonstrated that the anti-F2 fraction antibody, at different concentrations, was not effective in neutralizing the lethality of the total venom. Death of all animals was observed within 6 h after inoculation, confirming that in vivo protection was not achieved experimentally.

## 3. Discussion

The global mortality rate caused by snake envenomation is extremely high, especially in underdeveloped countries located in Africa, Asia, and Latin America [1]. *B. arietans*, responsible for most accidents with snakes on the African continent, is endemic to Sub-Saharan Africa but is also found in Morocco and Saudi Arabia. Its venom can cause local and systemic effects, such as pain, fever, leukocytosis, and cardiovascular and hypothermic disorders, which can culminate in the individual’s death [11,12].

According to the literature, toxins, such as hemorrhagic [32,33] and non-hemorrhagic [34] metalloproteases, fibrinogenolytic serine proteases, releasing kinins and acting on insulin [26], serine proteases with the ability to release kalidine [35], phospholipases A2, such as bitanarine [26,36] and Ba25, with a primary structure similar to venom type C lectins [37], PAL (Puff adder lectin), which induces the release of Ca^2+^ in the sarcoplasmic reticulum [38], and bitistatin (P17497), also known as arietin, a peptide capable of inhibiting platelet aggregation [39], were isolated from the venom of *B. arietans* (BaV). Contributing to hypotension is a family of ACE inhibitor peptides, the BPPs (Bradykinin-Potentiating Peptides), which have already been described [22], and high concentrations of adenosine, which can cause local vasodilation [40].

In this study, a group of representatives of one of the main classes of BaV toxins was isolated: metalloproteases (SVMPs) with masses ranging from 17 to 135 kDa. The molecular mass of the proteases found in the purified F2 fraction is similar to those found in the transcriptome and proteome of the venom [20,21]. Initially, the proteases were obtained through affinity column fractionation, which was confirmed by proteolytic activity and enzymatic inhibition tests. After confirming the presence of these proteases in the F2 fraction, a sample sent for analysis by mass spectrometry showed the presence of a greater abundance of SVMPs, specifically SVMPs of the PII class known as BA-5A. As has already been described in the literature, BA-5A, a disintegrin with action similar to SVMPs, has a conformation similar to class PII SVMPs, and it is suggested that BA-5A may represent an evolution in the evolutionary pathway of bitistatin [41], formerly known as arietin, a peptide capable of inhibiting platelet aggregation [21,39]. The low occurrence of BA-5A demonstration in the BaV proteome may mean that the structure of this protein is still poorly known. In addition to SVMPs, hyaluronidases were found. It was experimentally confirmed that this enzyme acts as a “spreading factor” in the toxic action of snake venom [27], due to its ability to hydrolyze hyaluronan and chondroitin sulfates A and C, facilitating the diffusion of toxins through tissues and blood circulation [27,42,43,44], contributing to local or systemic poisoning [42]. The degradation of hyaluronan from the extracellular matrix at the bite site contributes to tissue destruction, and the entry of venom hyaluronidase into the bloodstream degrades circulating hyaluronan, possibly causing systemic collapse [45]. Studies show that the enzyme was isolated from the venom of Angkistrodon acutus [43], Angkistrodon contortrix contortrix [46], Naja naja [47], and Daboia russelli [48]. Snaclecs found in BaV act together with hemorrhagic proteases, directly influencing hemostasis. It is described that snaclecs have platelet activation and aggregation properties similar to bitistatin from *B. arietans* and botrocetin from *Bothrops jararaca* [25,26].

The anti-F2 fraction antibodies were tested in ELISA titration assays against the venom of four specimens of snakes of the genus *Bitis*: *B. arietans*; *B. gabonica*; *B. nasicornis*; and *B. rhinoceros*. From the results obtained, it was verified that the antibodies do not present baseline levels of recognition to show cross recognition against the venoms of *B. gabonica*, *B. nasicornis*, and *B. rhinoceros*, corroborating the results obtained previously [19]. It was demonstrated in this study that *B. arietans*, as well as *B. caudalis*, do not establish relationships with other snakes of the species. BaV differs in that it contains large amounts of the disintegrin bitistatin [21], while the venom of other Bitis species contains dimeric disintegrins. BaV lacks C-type (αβ)3 lectin molecules and dimeric P-III class SVMPs, both conserved in *B. gabonica*, *B. nasicornis*, *B. rhinoceros*, and *B. caudalis* [19]. On the other hand, the results obtained with BaV demonstrated a high level of recognition against proteases present in the venom, validating the possible specificity of the antibodies.

Results of the affinity test of the anti-F2 fraction antibodies showed an affinity of 31% and 28% against the F2 fraction and the BaV, respectively, a percentage that represents the similarity of the antibodies bound between the groups. The inhibition test performed with BaV using the anti-F2 fraction antibodies revealed the inhibitory capacity of the antibodies against the proteases present in the venom. The results calculated as average reaction speed demonstrate that there was a reduction in the speed of action of the proteases, evidencing the efficiency of the antibodies against the venom toxins.

Complementing the in vitro characterization tests of the anti-F2 fraction antibodies, cross-reactivity analyses by immunoblotting show that the antibodies present a different recognition profile against the venoms of *B. gabonica*, *B. nasicornis*, and *B. rhinoceros*. It is possible to faintly observe the recognition of proteins of approximately 52 kDa in the venoms of the three snakes, in addition to those present in the range of approximately 50 and 26 kDa in the F2 fraction. Therefore, bands from F2 fraction and the recognized bands from *B. gabonica*, *B. nasicornis*, and *B. rhinoceros* venoms were submitted to gel digestion and sent to mass spectrometry, which demonstrated the absence of peptides, such as SVMPs and snaclecs in the bands of the Bitis venoms, justifying the weak recognition of the bands and correlating with the non-recognition of the venoms by the anti-F2 fraction of antibodies. On the other hand, the analysis of the bands corresponding to the F2 fraction demonstrated the presence of peptides from the main families that make up the venom, SVMPs, and snaclecs. It was also possible to observe the presence of peptides that matched the composition of the venom and the F2 fraction.

In order to analyze the hemorrhagic effects and the neutralizing capacity of anti-F2 fraction antibodies against the lethality and hemorrhagic action of BaV, the experimental models were used to verify the capacity of the venom and antibodies at different concentrations. From the evaluation of the hemorrhagic capacity of BaV, the minimum hemorrhagic dose (MHD) was calculated for the hemorrhagic serum neutralization assay, which demonstrated that the anti-F2 fraction antibodies were capable of reducing up to 80% of the hemorrhagic diameter in tissue fragments. The same neutralization was not observed in the lethality serum neutralization assay. In contrast, it has already been demonstrated that 77 µg of venom could be neutralized by 1 mg of antivenom produced in horses [49].

Based on the obtained results, we demonstrated through mass spectrometry the presence of peptides in F2 fraction, with the presence of SVMPs, hyaluronidases, and snaclecs being prevalent. The findings demonstrate SVMPs that have a similarity of practically 100% in all comparisons made by the databases with SVMPs present in viperid venom, such as *Bothrops jararaca*, *Bothrops atrox*, *Bothrops insularis*, and *Crotalus durissus durissus*. We observed the high immunogenic capacity of the anti-F2 fraction antibodies when performing the antibody titration during the immunization phase and the recognition of venom proteins by the antibodies in the ELISA titration test, immunoblotting, inhibition, and affinity tests. In addition, in vivo analyses showed different results regarding lethality and hemorrhage caused by the venom.

Regularly, polyvalent antivenoms use multispecific mixtures of venoms for immunization [50,51,52,53]. Snake venoms have in their composition different types of complex components with different immunogenic properties [18,20,21]. Antivenoms produced from snake venom toxins can cross-react in recognition against toxins present in other snake species used in the production of these antivenoms [54]. Faced with the production of a possible specific antibody against SVMPs belonging to the BaV, its addition to the polyvalent serum could be performed in a complementary way in order to increase the immunizing factor in victims of snakebites. By increasing the specific activity of the antivenom with the addition of a specific antibody against BaV SVMPs, lower doses of the antivenom will be required for the treatment of envenomations, reducing the amount and possibility of undesirable reactions. Compiling all the results obtained in this study, a more specific perspective on the components of the venom can show more complex responses to envenomation and treatment with antivenoms.

## 4. Conclusions

The BaV was fractionated by affinity chromatography, and one of the peaks resulting from the fractionation (fraction F2) was characterized. SVMPs, one of the main venom representatives of snakes in the Viperidae family, were the main focus of this work. The SVMPs found in the F2 fraction were partially characterized. Bleeding is one of the effects most associated with class P-I and P-III SVMPs. Whether they are hemorrhagic or not, SVMPs are directly involved in disorders of the coagulation cascade because they are fibrinogenolytic. Enzymatic analyses confirmed the predominant presence of SVMPs in the F2 fraction. Results obtained by MS/MS analysis demonstrated the presence of two enzymes that can influence envenoming—hyaluronidases and snaclecs. The presence of these enzymes can increase the occurrence of local and systemic effects by acting together with the venom proteases and increasing the diffusion of the toxic action of SVMPs. Regarding the production of antibodies, the purified F2 fraction of BaV used as antigen, together with the SBA-15 adjuvant, demonstrated great immunogenicity in the immunization cycle. Immunoassays performed showed that all mice had a high yield of anti-F2 fraction antibodies. The characterization of these antibodies demonstrated some important points for understanding the toxic action of snake venom and for the development of new alternative therapies: (1) snakes of the same genus have differences in the composition of the venom that can interfere in the treatment of victims, a fact observed in the cross-recognition assays and MS/MS analysis of *B. gabonica*, *B. nasicornis*, and *B. rhinoceros* venoms; (2) monospecific antibodies can serve as a treatment for local effects, as demonstrated in the *B. arietans* venom hemorrhagic activity inhibition test; (3) as they are monospecific, these antibodies are not able to neutralize the lethality, but complement the treatment by neutralizing the local effects; and (4) the addition of specific local treatments can help with antivenom therapy and decrease mortality and disabilities caused by envenoming.

This work presents original results that make possible the understanding of the action of the snake venom and also the principles to establish an improvement in the treatments with polyvalent antivenoms through the addition of possible specific antibodies against several actions of the venom. Based on this, we conclude that the results obtained contribute to the treatment of poisoned people and, consequently, to the improvement of the anti-Bitis serum production.

## 5. Materials and Methods

### 5.1. Venom

Lyophilized Bitis arietans venom (BaV) was purchased from the Venom supplies (South Australia, Australia), obtained from adult male and female animals, measuring approximately 60 cm each, and coming from the south of the African continent. Stock solutions were prepared in sterile phosphate-buffered saline (PBS, 8.1 mM sodium phosphate, 1.5 mM potassium phosphate, 137 mM sodium chloride, and 2.7 mM potassium chloride, pH 7.2) at 1 mg/mL based on their protein concentration measured by the bicinchoninic acid method [30] using a Pierce BCA Protein Assay kit (Rockford, IL, USA), with bovine serum albumin as the standard protein, and stored in a −20 °C freezer.

### 5.2. Isolation and Identification of Venom Toxins by Affinity Chromatography

The venom was fractionated by affinity chromatography on an immobilized zinc column (Hitrap IMAC HP 1 mL) equilibrated with TRIS 25 mM, pH 8. Protein content was determined using the UPC-900 reader and EKTApurifier system (Pharmacia Biotech AB, Amersham, ENG) under absorbance at 280 nm. Briefly, 60 mg of the venom was diluted in buffer A (TRIS 25 mM, pH 8) and injected into the column. Proteins were eluted at a rate of 1 mL/min in buffer B (100 mM Imidazole; 20 mM TRIS pH 8) and 2 mL of the peak identified as F2 fraction was collected. The protein content was determined using the Pierce BCA Protein Assay kit (Pierce Biotechnology, Rockford, IL, USA) according to the manufacturer’s instructions [30]. For data intercalation, a standard curve was used with concentrations of 0–2000 µg/mL of fetal bovine serum albumin (BSA, Sigma–Aldrich, St. Louis, MO, USA) diluted in PBS pH 7.2. The reading was taken with ELX 800 plate spectrophotometer (Biotek Instruments, Winooski, VT, USA) at a wavelength of 540 nm. The absorbance values of the samples were intercalated with the values of the standard curve, and the concentration of the samples was determined. The electrophoretic profile was visualized using SDS-PAGE and stained with silver. Samples of F2 fraction (2.5 µg) and BaV (2 µg) were submitted to polyacrylamide gel electrophoresis [55,56]. The amounts of 5% and 12% acrylamide gels were used for the top gel and bottom gel, respectively. For the development of the assay, the samples were diluted in non-reducing buffer in a ratio of 1:1 and applied to the gel in individual wells, where they were subjected to a constant current of 100V. Silver impregnation revealed the gels [56]. 

### 5.3. Proteolytic Activity of the F2 Fraction on FRET Substrates

The proteinase activity assays were conducted in PBS pH 7.2 (final volume 100 μL) using 96-well plates (Perkin Elmer, Waltham, MA, USA), the Fluorescent Resonance Energy Transfer (FRET) substrates Abz-RPPGFRSPFR-QEDDnp, already defined as selective for the SVMP present in the BaV [57], and the substrate Abz-FRSSF-EDDnp, already defined for SVSP [14], both provided by Dra. Fernanda Calheta Vieira Portaro, Instituto Butantan, Laboratory of Structure and Function of Biomolecules.

At a concentration of 250 mM in PBS pH 7.2 (final volume 100 µL/well), fractionated F2 and FRET substrates were used. The tests were performed in a thermostabilized compartment (37 °C) under agitation and the reactions were continuously monitored (fluorescence emission at 420 nm after excitation at 320 nm) in a fluorimeter (Hidex Sense, Hidex, Finland). Proteolytic activity was expressed as mean reaction velocity (Vm). The experiments were performed in duplicate.

### 5.4. Inhibition of the Proteolytic Activity of the F2 Fraction with Selective Inhibitors

The proteolytic inhibition assay started with the incubation of the samples for 30 min at room temperature with the active site inhibitor phenylmethylsulfonyl (PMSF, 2 mM/well). Ethylenediamine tetraacetic acid (EDTA, 100 mM/well) was used without pre-incubation time, being added directly to the assay. 

Control samples were also prepared in the same volumes, where the BaV, enzymatic inhibitor, and substrate at 250 µM in PBS pH 7.2 (totaling 100 µL) were applied for the positive control, and for the negative control, the substrate at 250 µM and enzyme inhibitor was added to PBS pH 7.2 (totaling 100 µL). The tests were performed in a thermostabilized compartment (37 °C) under agitation, and the reactions were continuously monitored (fluorescence emission at 420 nm after excitation at 320 nm) in a fluorimeter (Hidez Sensa, Hidex, Finland). The Abz—RPPGFRSPFR-QEDDnp substrate (250 mM) was used, since this FRET peptide, in the experiments described above, was the most selective for *B. arietans* SVSPs. Inhibition of proteolytic activity was expressed as mean reaction velocity (Vm). The experiments were performed in duplicate.

### 5.5. Mass Spectrometry of the F2 Fraction

The purified F2 fraction from affinity chromatography was submitted to digestion with trypsin (Sigma–Aldrich, Missouri, USA) in the solution. The solution was desalted by Zip-Tip (C18), dried, and resuspended in 0.1% formic acid and sent for analysis in a mass spectrometer (LC-MS/MS) at the CENTD Special Laboratory of the Butantan Institute. The analysis was carried out using liquid chromatography on an Easy-nLC Proxeos nanoHPLC system, which was linked to an LTQ-Orbitrap Velos (Thermo Fisher Scientific, Bremen, Germany) via a nanoelectrospray ionization source. The peptides were separated on a C18 column (Jupiter^®^, Phenomenex, Torrance, CA, USA) and packed in-house with 5 µm beads. The peptides were then eluted in a linear gradient of 5–95% acetonitrile in 0.1% formic acid for 15 min at a continuous flow rate of 200 nL/min. MS spectra were obtained by Orbitrap (scan range: 200–2000 m/z; full scan resolution: 60,000; maximum injection time: 100 ms). Nanoelectrospray voltage was set to 2.1 kV, and a source temperature of 200 C, and the spectrometer operated in data-dependent mode (DDA), where the 5 most intense peaks were selected for collision-induced dissociation(CID) fragmentation (2 D isolation window; exclusion time 15 s; minimum signal 5000; activation time = 10 ms; normalized collision energy = 35%). The data files were submitted to a search against the “Serpentes” database (taxid: 8570) using BLASTP 2.13.0 [31] UniProtKB/SwissProt and PEAKS DB.

### 5.6. Production of Anti-F2 Fraction Antibodies

For the production of the anti-F2 fraction, isogenic female mice of the BALB/c strain weighing 18–22 g were used. The animals (n = 10) were inoculated with 0.5 µg/animal of the F2 fraction; the dose was calculated according to the LD-50 already established by our group [49]. For this purpose, the animals were submitted to five subcutaneous inoculations with a F2 fraction diluted in 0.15 M non-pyrogenic saline solution with nanostructured mesoporous silica (SBA-15) as an adjuvant in a 10:1 ratio, which was administered in a volume of 500 µL, at 15-day intervals. Approximately 50–100 µL of blood from each mouse was collected every 15 days by mandibular puncture and placed in an Eppendorf tube containing an anticoagulant solution (Heparin 1000 U/mL). After homogenization, the blood was centrifuged at 1620× *g* for 45 min at 4 °C to obtain plasma, heated to 56 °C for 30 min to inactivate the complement system, and stored in a freezer at −20 °C for further analysis. Following the immunization cycle, the mice were sacrificed intraperitoneally with an overdose of the anesthetics Xylazine (30 mg/kg) and Ketamine (300 mg/kg). 

All animals used in this study come from the Central Animal Facility of the Instituto Butantan were kept in the experimental animal facility of the Immunochemistry Laboratory under standardized conditions, with water and food available ad libitum, in accordance with the certificate approved by Ethics Committee on the Use of Animals at the Butantan Institute (CEUA IB) (no. 3874060420) and norms established by National Board of Control of Animal Experimentation (CONCEA).

### 5.7. Quantification of Anti-F2 Fraction Antibodies 

High-affinity 96-well ELISA plates (COSTAR^®^ 3590) were incubated overnight at 4 °C with 1 µg/well of the F2 fraction in 100 µL of PBS pH 7.2. The plates were blocked for 2 h at 37 °C with 200 µL/well of PBS/BSA 5%. After blocking for 2 h at 37 °C with 200 µL/well of PBS/BSA 5%, the plates were washed briefly with 200 µL/well of PBS, pH 7.2. Serial dilutions of mouse plasma (1:20 to 1:10,240) in PBS/BSA 0.1% were prepared, and 100 µL/well of each dilution was added into their respective wells. The plates were then incubated at 37 °C for 1 h, and washed three times with 200 µL/well of wash buffer (PBS/Tween-20 0.05%). Goat peroxidase-conjugated anti-mouse IgG (Sigma–Aldrich, St. Louis, MO, USA) diluted (1:7500) in PBS/BSA 0.1% (100 µL/well) was added, and the plates were incubated for 1 h at 37 °C. After three washes with wash buffer, 50 µL/well of substrate buffer for ELISA was added, and the plates were incubated at room temperature in the dark for 20 min. The reaction was terminated by adding 50 µL/well of 2N sulfuric acid. The reading was taken with ELX 800 plate spectrophotometer (Biotek Instruments, Vermont, USA) at a wavelength of 490 nm. Wells without the antigen and wells without the primary antibody were used as controls. The antivenom dilution with an optical density (OD) of 0.2 was used to calculate the U-ELISA per milliliter of the antivenom solution. A U-ELISA was defined as the lowest antibody dilution that presented an OD of 0.2 under the conditions used in the assay, as previously described [58]. End Point 3x the controls (U/mL) or ELISA-units/mL (EU/mL) were used to represent the titration. The value was multiplied by 10 to convert it to milliliters.

### 5.8. Quantification of Anti-F2 Fraction Antibodies against Venoms of Snakes of the Genus Bitis

The quantification of the anti-F2 fraction antibodies obtained was determined through the ELISA assay against the venoms of the snakes *B. arietans*, *B. gabonica*, *B. nasicornis*, and *B. rhinoceros* (Venom Supplies, South Australia, Au). Briefly, high-affinity 96-well ELISA plates (COSTAR^®^ 3590) were incubated overnight at 4 °C with 1 µg/well of the *B. gabonica*, *B. nasicornis*, or *B. rhinoceros* venom in 100 µL of PBS pH 7.2 buffer and blocked for 2 h at 37 °C with 200 µL/well of PBS/BSA 5%. Serial dilutions of mouse final bleed plasmas (1:500 to 256,000) in PBS/BSA 0.1% were prepared, and 100 µL/well of each dilution was added to their respective wells. The plates were incubated at 37 °C for 1 h and then washed three times with 200 µL/well of wash buffer (PBS/Tween-20 0.05%). Goat peroxidase-conjugated anti-mouse IgG (Sigma–Aldrich, St. Louis, MO, USA) diluted (1:5000) in PBS/BSA 0.1% (100 µL/well) was added, and the plate was incubated for 1h at 37 °C. After three washes with wash buffer, 50 µL/well of substrate buffer for ELISA was added, and the plate was incubated at room temperature, protected from light, for 20 min. The reaction was stopped by adding 50 µL/well of 2N sulfuric acid. The reading was taken with ELX 800 plate spectrophotometer (Biotek Instruments, Vermont, USA) at a wavelength of 490 nm. Wells without the antigen and primary antibody were used as controls. Titer was presented as a 0.3x the control (U/mL) or ELISA-units/mL (EU/mL) endpoint. The value was multiplied by 10 to convert it to milliliters. Data were statistically analyzed using GraphPad Prism version 7 for Windows (GraphPad Software, San Diego, CA, USA).

### 5.9. Determination of the Affinity of Anti-F2 Fraction Antibodies 

The determination of the affinity of the anti-F2 fraction antibodies was carried out through the ELISA method, as previously described, with some modifications. The anti-F2 fraction antibodies were used as primary antibodies at a fixed dilution of 1:1000 (value determined from the analysis by ELISA of the results referring to the yield of the antibodies against the BaV). After incubation of the primary antibody, potassium thiocyanate (KSCN, Sigma–Aldrich, St. Louis, Missouri, USA) was added as a chaotropic agent [59,60]. KSCN was prepared in dilutions from 0 M to 5 M, with intervals of 1 M, in distilled water. An amount of 150 µL/well of each KSCN dilution was applied. The plates were incubated at room temperature for 30 min and then washed three times with 200 µL/well of washing buffer (PBS/Tween-20 0.05%). Thereafter, the reaction proceeded, as described above. The reading was taken with ELX 800 plate spectrophotometer (Biotek Instruments, Vermont, USA) at a wavelength of 490 nm. The affinity value was determined as the concentration of KSCN necessary to reduce the optical density obtained with 0 M KSCN by 50% [61]. Data were statistically analyzed using GraphPad Prism version 7 for Windows (GraphPad Software, San Diego, CA, USA).

### 5.10. Immunoblotting of the F2 Fraction and Venoms of Snakes of the Genus Bitis against Anti-F2 Fraction Antibodies

The venom of *B. arietans*, *B. gabonica*, *B. nasicornis*, and *B. rhinoceros* (2 µg/well), in addition to the F2 fraction (2,5 µg/well), was submitted to polyacrylamide gel electrophoresis; 5% and 12% acrylamide gels were used for the top gel and bottom gel, respectively, under 100 V constant current. For the development of the assay, the samples were diluted in non-reducing buffer in a 1:1 ratio and applied to the gel in individual wells. Then, the protein bands were transferred to 0.45 µm nitrocellulose membranes (Trans-Blot, Bio-Rad, DE) in a TE-22 transfer vessel (Hoefer Pharmacia Biotech, San Francisco, CA, USA) overnight at 150 mA. After transfer, the non-specific sites of the membranes were blocked in PBS/BSA 5% for 2 h at 37 °C. Afterward, they were washed quickly with PBS pH 7.2 and incubated with the anti-F2 fraction antibodies at a 1:200 dilution in PBS/BSA 0.1% for 1 h at 37 °C. After incubation, the membranes were washed three times with PBS/Tween-20 0.05%. Then the membranes were incubated with the secondary antibody, goat anti-mouse IgG conjugated with alkaline phosphatase (Sigma–Aldrich, St. Louis, MO, USA), diluted 1:5000 in PBS/BSA 0.1% for 1 h at 37 °C, and then washed three times with PBS/Tween-20 0.05% and placed for development in NBT/BCIP developer solution (Sigma–Aldrich, St. Louis, MO, USA). The development of the bands was followed visually, and the reaction was ended by placing the membranes in distilled water. 

### 5.11. Mass Spectrometry of F2 Fraction Bands and Venoms from Snakes of the Genus Bitis

Bands from electrophoresis performed in 4.10 were selected and submitted to in-gel digestion with trypsin (Sigma–Aldrich, Missouri, EUA). The bands were desalted by “Zip-Tip” (C18), dried and resuspended in 0.1% formic acid, and sent for analysis via a mass spectrometer (LC-MS/MS) at the Special Laboratory of Applied Toxinology (LETA) at Instituto Butantan. The analysis was carried out using liquid chromatography on an Easy-nLC Proxeos nanoHPLC system, which was linked to an LTQ-Orbitrap Velos (Thermo Fisher Scientific, Bremen, Germany) through a nanoelectrospray ion source. The peptides were separated on a C18 column (Jupiter^®^, Phenomenex, Torrance, CA, USA) and packaged “in-house” with 5 µm beads. The peptides were then eluted in a linear gradient of 5–95% acetonitrile in 0.1% formic acid for 15 min at a continuous flow rate of 200 nL/min. MS spectra were obtained by Orbitrap (scan range: 200–2000 m/z; full-scan resolution: 60,000; maximum injection time: 100 ms). The nanoelectrospray voltage was set to 2.1 kV, the source temperature to 200 °C, and the spectrometer operated in data-dependent mode (DDA) where the 5 most intense peaks were selected for collision-induced dissociation (CID) fragmentation (2 D isolation window; 15 s exclusion time; minimum signal 5000; activation time = 10 ms; normalized collision energy = 35%). The data files were submitted to a search against the “Serpentes” database (taxid: 8570) using BLASTP 2.13.0 [31], UniProtKB/SwissProt, and PEAKS DB.

### 5.12. Inhibition of the Proteolytic Activity of BaV and the F2 Fractions with Anti-F2 Fraction Antibodies

The proteolytic inhibition assay started with the incubation of the samples for 30 min at room temperature with the anti-F2 fraction antibodies. Control samples were also prepared in the same volumes, where the BaV or F2 fraction and the Abz-RPPGFRSPFR-QEDDnp substrate at 244 µM in PBS pH 7.2 (totaling 100 µL) were applied for the positive control. The tests were performed in a thermostabilized compartment (37 °C) under agitation, and the reactions were continuously monitored (fluorescence emission at 420 nm after excitation at 320 nm) by fluorimeter (Hidex Sensa, Hidex, Finland). The experiments were performed in duplicate.

### 5.13. Evaluation of the Hemorrhagic Activity of BaV

For the evaluation of the hemorrhagic activity of BaV, heterogeneous mice of the Swiss strain weighing 18–22 g were used. The animals (n = 20) were inoculated with increasing doses of venom and were prepared in PBS pH 7.2 (0, 10, 20, 30, and 40 µg/animal). Control samples were also prepared. The animals were injected with 50 µL of each preparation intradermally. After 3 h, the animals were sacrificed intraperitoneally with an overdose of the anesthetics Xylazine (30 mg/kg) and Ketamine (300 mg/kg), the skin was dissected, and the hemorrhagic activity was calculated by measuring the diameter of the hemorrhagic halo with the aid of the ImageJ 1.8.0 program [62,63], and presented as square millimeters (mm^2^).

All animals used in this study came from the Central Animal Facility of the Instituto Butantan and were kept in the experimental animal facility of the Immunochemistry Laboratory under standardized conditions, with water and food available ad libitum, in accordance with the certificate approved by the Ethics Committee on the Use of Animals at the Butantan Institute (CEUA IB) (no. 9426250422) and norms established by National Board of Control of Animal Experimentation (CONCEA).

### 5.14. In Vivo Hemorrhage Neutralization

For the test, heterogeneous mice of the Swiss strain, weighing 18–22 g, were used. The animals (n = 16) were inoculated with 10 µg/animal of BaV, and the value was determined as minimum hemorrhagic dose (DHM) diluted in PBS pH 7.2. This value was calculated from the results obtained in the evaluation of the hemorrhagic activity of the venom. Increasing preparations of antibodies (1:5, 1:10, and 1:20) diluted in PBS pH 7.2 were added to the venom. The preparations were incubated for 30 min at room temperature under agitation and then inoculated into the animals. Venom samples prepared only with PBS pH 7.2 were injected into the control group. The animals were injected with 50 µL of each preparation intradermally.

All animals used in this study came from the Central Animal Facility of the Instituto Butantan and were kept in the experimental animal facility of the Immunochemistry Laboratory under standardized conditions, with water and food available ad libitum, in accordance with the certificate approved by the Ethics Committee on the Use of Animals at the Butantan Institute (CEUA IB) (no. 9426250422) and norms established by National Board of Control of Animal Experimentation (CONCEA).

### 5.15. In Vivo Lethality Neutralization

For the assay, heterogeneous mice of the Swiss strain, weighing 18–22 g, were used. The animals (n = 16) were inoculated with the amount of BaV corresponding to 2 LD50 (100 µg/animal) diluted in PBS pH 7.2. Increasing antibody preparations were added to the venom (1:10, 1:20, and 1:40) diluted in PBS pH 7.2. The preparations were incubated for 30 min at room temperature under agitation and then inoculated into the animals. Venom samples prepared only with PBS pH 7.2 were injected into the control group. The animals were injected with 500 µL of each preparation intraperitoneally. The death to survival ratio was recorded 3 h, 24 h, and 48 h after the inoculations, and the effective dose (ED50) was determined by probits [64].

All animals used in this study came from the Central Animal Facility of the Instituto Butantan and were kept in the experimental animal facility of the Immunochemistry Laboratory under standardized conditions, with water and food available ad libitum, in accordance with the certificate approved by the Ethics Committee on the Use of Animals at the Butantan Institute (CEUA IB) (no. 9426250422) and norms established by National Board of Control of Animal Experimentation (CONCEA).

### 5.16. Statistical Analysis

The data were expressed as mean ± standard deviation and analyzed statistically using the software GraphPad Prism, version 7 for Windows (GraphPad Software, San Diego, CA, USA). The statistical significance of results was calculated by the One-way analysis of variance (ANOVA) test followed by Dunnet’s Post Test. Differences were considered to be significant if *p* < 0.05.

## Figures and Tables

**Figure 1 toxins-15-00264-f001:**
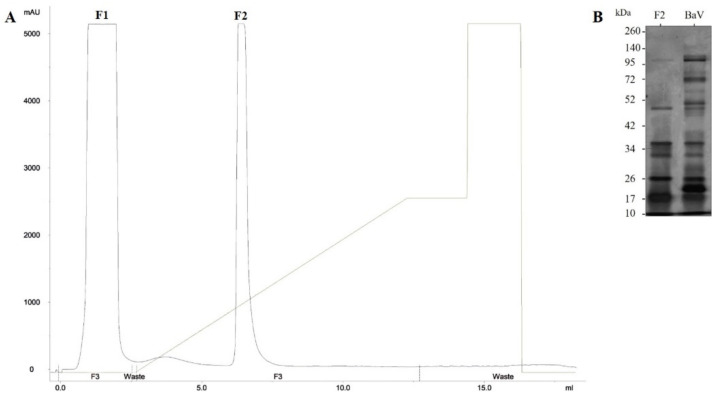
BaV chromatographic profile and electrophoretic profiles of F2 fraction and the BaV. (**A**) Sixty milligrams of lyophilized venom were subjected to a chromatography step on an affinity column, equilibrated, and eluted with 25 mM TRIS, pH 8. Samples were eluted at a continuous flow rate of 1 mL/min, and the protein content was monitored under absorbance at 280 nm on the UPC-900 reader. (**B**) Five µg of BaV and five µg of the F2 fraction were subjected to gradient gel electrophoresis (5% for the upper gel and 12% for the lower gel) under non-reducing conditions. The bands were revealed by silver nitrate impregnation.

**Figure 2 toxins-15-00264-f002:**
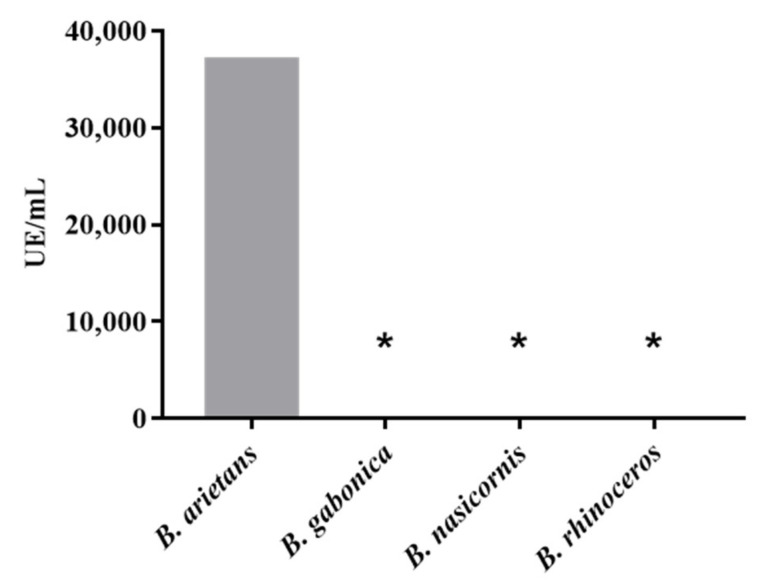
Cross-recognition by ELISA of the venoms of snakes of the genus *Bitis* against the F2 anti-fraction antibody. Cross-recognition of antibodies against *B. arietans*, *B. gabonica*, *B. nasicornis* and *B. rhinoceros* venoms was determined by the ELISA method in 96-well plates sensitized with one µg of antigen/well. The anti-F2 fraction antibodies were serially diluted (1:500 to 1:256.000) in PBS/BSA 0.1%. Detection with peroxidase-conjugated “anti-mouse” antibodies was performed at a dilution of 1:5000. The reading was taken with ELX 800 plate spectrophotometer (Biotek Instruments, Vermont, USA) at a wavelength of 490 nm. The yield was presented as ELISA units/mL (EU/mL). The assay was performed in duplicate. Data were statistically analyzed using GraphPad Prism version 7 for Windows (GraphPad Software, San Diego, CA, USA). The yield was presented as units per milliliter. * *p* < 0.05.

**Figure 3 toxins-15-00264-f003:**
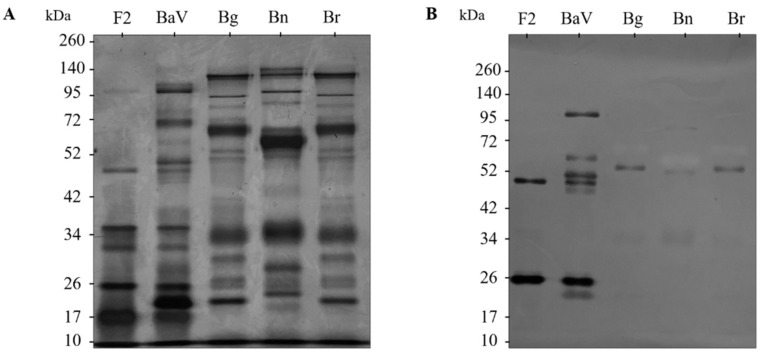
Cross-recognition by immunoblotting of venoms from snakes of the genus Bitis against the anti-F2 fraction antibody. (**A**) Five µg of each venom and five µg of the F2 fraction were subjected to gradient gel electrophoresis (5% for upper gel and 12% for lower gel) under non-reducing conditions. The bands were revealed by silver nitrate impregnation. (**B**) The nitrocellulose membrane was incubated for 1 h at room temperature with the anti-F2 fraction antibody diluted 1:200 in PBS/BSA 0.1%. After washing with PBS/Tween-20 0.5%, the membrane was incubated for 1 h at room temperature with the “anti-mouse” IgG antibody conjugated with alkaline phosphatase, diluted 1:5000 in PBS/BSA 0.1%.

**Figure 4 toxins-15-00264-f004:**
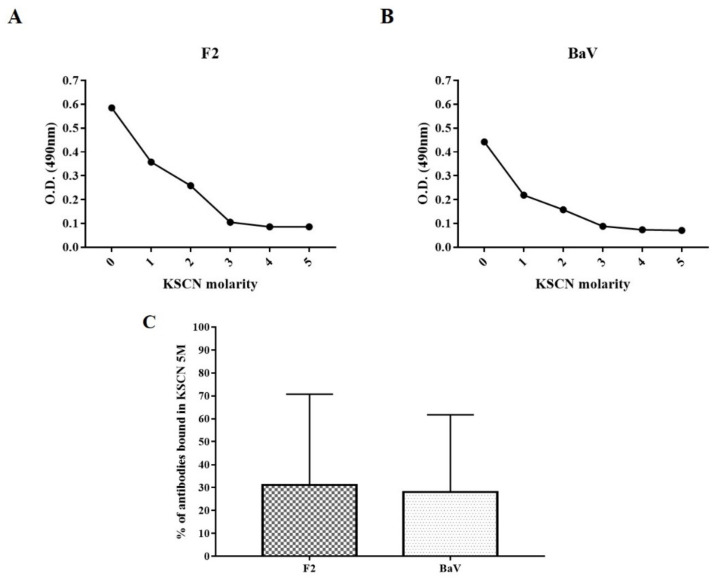
Experimental plasma affinity. A 96-well plate was primed with one µg of antigen/well, and the dilution of anti-F2 fraction antibodies was set at 1:1000. KSCN concentration ranged from 0 M to 5 M. The percentage of antibodies bound to 3 M KSCN was used to calculate affinity. (**A**) F2 affinity curve. (**B**) raw venom affinity curve. (**C**) percentage of antibodies bound to KSCN 5M. The assay was performed in duplicate. Data were statistically analyzed using GraphPad Prism version 7 for Windows (GraphPad Software, San Diego, CA, USA). The yield was presented as units per milliliter.

**Figure 5 toxins-15-00264-f005:**
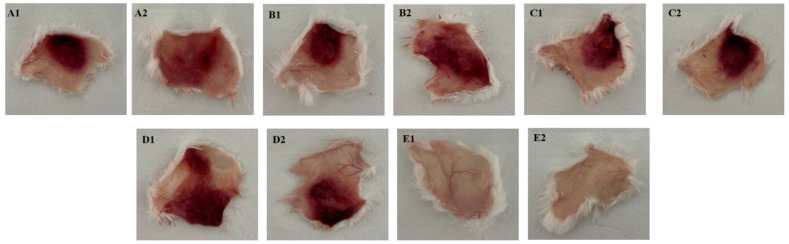
Hemorrhagic activity of BaV. Groups of mice (n = 20) were inoculated by intradermal injection with increasing amounts of BaV or PBS pH 7.2. (**A**) Hemorrhagic tissue fragments corresponding to 10 µg/animal. (**B**) Hemorrhagic tissue fragments corresponding to 20 µg/animal. (**C**) Hemorrhagic tissue fragments corresponding to 30 µg/animal. (**D**) Fragments of hemorrhagic tissue corresponding to 40 µg/animal. (**E**) Fragments of hemorrhagic tissue corresponding to PBS pH 7.2 inoculation in control animals. The diameter of the area of each tissue fragment was plotted in the ImageJ 1.8.0 program and expressed in mm^2^.

**Figure 6 toxins-15-00264-f006:**
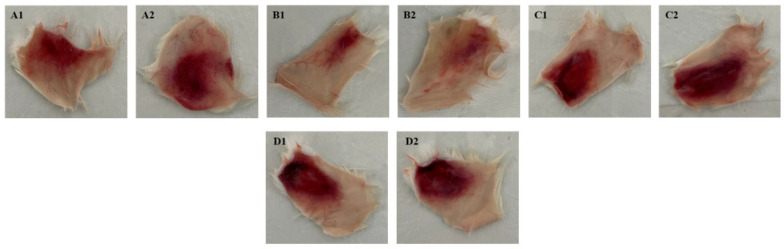
Serum neutralization of BaV hemorrhagic activity. Groups of mice (n = 16) were inoculated by intradermal injection with the MHD of 10 µg of BaV together with different concentrations of the anti-F2 fraction antibody, or PBS pH 7.2. (**A**) Hemorrhagic tissue fragments corresponding to 10 µg/animal + anti-F2 fraction antibodies 1:5. (**B**) Hemorrhagic tissue fragments corresponding to 10 µg/animal + antibodies anti-F2 fraction 1:10. (**C**) Hemorrhagic tissue fragments corresponding to 10 µg/animal + antibodies anti-F2 fraction 1:20. (**D**) Hemorrhagic tissue fragments corresponding to the inoculation of 10 µg/animal of BaV in PBS pH 7.2 in control animals. The diameter of the area of each tissue fragment was plotted in the ImageJ 1.8.0 program and expressed in mm^2^.

**Table 1 toxins-15-00264-t001:** Proteolytic activity of the F2 fraction.

Sample + Substrate	Vm*
BaV + Abz-RPPGFRSPFR-QEDDnp	6600
F2 fraction + Abz-RPPGFRSPFR-QEDDnp	6581
BaV + Abz-FRSSF-EDDnp	2430
F2 fraction + Abz-FRSSF-EDDnp	434

In white-backed 96-well plates, two µL/well of each sample was incubated with two µL/well of FRET substrates (Abz-RPPGFRSPFR-QEDDnp and Abz-FRSSF-EDDnp) in PBS pH 7.2 (final volume 100 µL/well). Proteolytic activity was measured by spectrofluorimetry (λEX = 320 and λEM = 420 nm, reader: Hidex Sense, Hidex, Finland) for 15 min. Representative data from two independent trials. The assay was performed in duplicate. Vm*: mean reaction rate—concentration variation/time interval.

**Table 2 toxins-15-00264-t002:** Mean rate of reaction of proteolytic inhibition of crude venom of *B. arietans* and the F2 fraction with specific inhibitors.

Sample + Inhibitor + Substrate	Vm*	Statistical Difference
BaV + PBS + Abz-RPPGFRSPFR-QEDDnp	9083	ns
BaV+ PMSF + Abz-RPPGFRSPFR-QEDDnp	7317	ns
BaV+ EDTA + Abz-RPPGFRSPFR-QEDDnp	793	***
F2 fraction + PBS + Abz-RPPGFRSPFR-QEDDnp	7500	ns
F2 fraction + PMSF + Abz-RPPGFRSPFR-QEDDnp	8280	ns
F2 fraction + EDTA + Abz-RPPGFRSPFR-QEDDnp	1820	***

In white-backed 96-well plates, two µL/well of the crude venom and the F2 fraction were applied with two µL/well of EDTA (100 mM), PMSF (2 mM) and PBS pH 7.2. After 30 min of incubation at room temperature, two µL/well of FRET Abz-RPPGFRSPFR-QEDDnp 250 µM substrate was added for a total volume of 100 µL/well. Proteolytic activity was measured by spectrofluorimetry (λEX = 320 and λEM = 420 nm, reader: Hidex Sense, Hidex, Finland) for 15 min. The assay was performed in duplicate. Representative data from two independent tests expressed as mean reaction rate (Vm* = mean reaction rate—concentration range/time interval) and analyzed by One-way ANOVA test followed by Dunnet’s Post Test. *p* < 0.001 (***). ns: not significant.

**Table 3 toxins-15-00264-t003:** Peptides identified in the F2 fraction.

	Sequence ^a^	*Bitis arietans* Peptide	Origin ^b^	Peptide
1	SDLLNRKRHD	BA-5A—(SVMP-PII)	*Echis coloratus*	Ecarin—(SVMP)
2	ELPKGAVQP	-	*Bothrops asper*	BaP1—(SVMP)
3	WRATDLLR	BA-5A—(SVMP-PII)	*Crotalus adamanteus*	SVMP
4	AENTLHSF	-	*Crotalus durissus terificus*	Atrolysin-B (mpo-1)—(SVMP)
5	GLARVGSMC	-	*Crotalus atrox*	MDC-8c—(SVMP)
6	LSGAKGNNC	-	*Bitis gabonica*	Snaclec 1
7	PQDWLPM	Snaclec PAL	*Bitis gabonica*	Snaclec 1
8	HGMVLPGT	-	*Bitis gabonica*	Bitisgabonin—(SVMP)
9	ITSPNVTLP	-	*Bitis gabonica*	SVSP 1
10	SGHLDSWDVGQGAMDDGGGA	Snaclec bitiscetin alfa	*Bitis gabonica*	Snaclec bitisgabonin 2
11	GHLDSWDVGQGAMDDGGGA	-	*Bitis gabonica*	Snaclec bitisgabonin 2
12	VGQGAMDDGGGAFIS	-	*Bitis gabonica*	Snaclec bitisgabonin 2
13	AENTLHSF	-	*Crotalus molossus*	SVMP
14	DHSPINL	-	*Crotalus atrox*	Atrolysin-A—(SVMP)
15	YEPIKKASDL	BA-5A—(SVMP-PII)	*Bitis gabonica*	SVMP
16	SDLLNRKRHD	BA-5A—(SVMP-PII)	*Bitis gabonica*	SVMP
17	ELPKGAVQP	BA-5A—(SVMP-PII)	*Bothrops asper*	BaP1—(SVMP-PI)
18	WRATDLLR	BA-5A—(SVMP-PII)	*Crotalus adamanteus*	SVMP
19	DHSSVNRL	-	*Crotalus durissus terificus*	Snaclec crotocetin-1
20	QDFSPIN	-	*Bothrops alternatus*	BaG—(SVMP)
21	KMFYSNDDEHKG	-	*Bothrops jararaca*	Bothropasin/jararagina—(SVMP)
22	ITSPNVTLP	-	*Bitis gabonica*	SVSP
23	LSGAKGNNC	-	*Bitis gabonica*	Snaclec 1
24	PQDWLPM	Snaclec PAL	*Bitis gabonica*	Snaclec 1
25	J3SBQ3	-	*Bitis gabonica*	SVSP
26	HGMVLPGT	-	*Bitis gabonica*	Bitisgabonin—(SVMP)
27	HLLTRKKHDNAQLLT	-	*Bitis gabonica*	Bitisgabonin—(SVMP)
28	DIVVSENI	Snaclec bitiscetin alfa	*Echis carinatus*	Snaclec echicetin alfa
29	NSLIDALMH	-	*Bothrops jararaca*	PLI alpha (PLA2 inhibitor)
30	MNSKC	-	*Notechis scutatus scutatus (Elapidae)*	Cysteine-rich venom protein
31	VNKTDKK	-	*Dendroaspis angusticeps (Elapidae)*	AdTx1/Muscarinic toxin 3
32	LSNLDRSHPW	BA-5A—(SVMP-PII)	*Bothrops jararaca*	Bothrojarin-3—(SVMP)
33	AMPAKAPMYPN	Hyaluronidase 1	*Bitis arietans*	Hyaluronidase 1
34	RDTLLLAEEMRPNGYW	Hyaluronidase 1 e 2	*Bitis arietans*	Hyaluronidase 1 e 2
35	EDLVTTVGETAAM	Hyaluronidase 1 e 2	*Bitis arietans*	Hyaluronidase 1 e 2
36	RDTLLLAEEMRPNGYWGYYLY	Hyaluronidase 1 e 2	*Bitis arietans*	Hyaluronidase 1 e 2
37	CQNYDYKTKGDQYTG	Hyaluronidase 1 e 2	*Bitis arietans*	Hyaluronidase 1 e 2
38	LFPESFRIM	Hyaluronidase 1 e 2	*Bitis arietans*	Hyaluronidase 1 e 2
39	VHANATEKK	Hyaluronidase 1 e 2	*Bitis arietans*	Hyaluronidase 1 e 2
40	TKHLNKSKSDI	Hyaluronidase 1 e 2	*Bitis arietans*	Hyaluronidase 1 e 2
41	GIVFWGSMQYASTVDSC	Hyaluronidase 1 e 2	*Bitis arietans*	Hyaluronidase 1 e 2
42	NRSIQFAKE	Hyaluronidase 1 e 2	*Bitis arietans*	Hyaluronidase 1 e 2
43	EEMRPNGYWGYY	Hyaluronidase 1 e 2	*Bitis arietans*	Hyaluronidase 1 e 2
44	EKAAKSFMRDTLLLAE	Hyaluronidase 1 e 2	*Bitis arietans*	Hyaluronidase 1 e 2
45	KHSDSNAFLHL	Snaclec PAL	*Bitis arietans*	Hyaluronidase 1 e 2
46	CPDIEM	Hyaluronidase 1 e 2	*Bitis arietans*	Hyaluronidase 1 e 2
47	LDLKTFHIV	Hyaluronidase 1 e 2	*Bitis arietans*	Hyaluronidase 1 e 2
48	PMYPNEPFIVLWN	Hyaluronidase 1 e 2	*Bitis arietans*	Hyaluronidase 1 e 2
49	QYASTVDSC	Hyaluronidase 1 e 2	*Bitis arietans*	Hyaluronidase 1 e 2
50	QKVKTYMNGPL	BA-5A—(SVMP-PII)	*Bitis arietans*	Hyaluronidase 1 e 2
51	AMPAKAPMYPN	Hyaluronidase 1 e 2	*Bitis arietans*	Hyaluronidase 1 e 2
52	RDTLLLAEEMRPNGYW	Hyaluronidase 1 e 2	*Bitis arietans*	Hyaluronidase 1 e 2
53	TASISAC	BA-5A—(SVMP-PII)	*Bitis arietans*	BA-5A—(SVMP-PII)
54	SELSVGLVQDYMP	BA-5A—(SVMP-PII)	*Bitis arietans*	BA-5A—(SVMP-PII)
55	IWMGLNDVWN	Snaclec 8	*Bitis arietans*	Snaclec 8
56	THNFVC	Snaclec 6	*Bitis arietans*	Snaclec 6
57	PEEGEREPSSPLTPGSL	Barietin—(VEGF-Fs)	*Bitis arietans*	Barietin—(VEGF-Fs)

The peptide sequences of fragments found in the F2 fraction, analyzed by mass spectrometry (LC-MS/MS), were searched in the UniprotKB/Swiss-Prot Serpentes database (taxid: 8570). ^a^ the According to PEAKS DB; ^b^ referring to the similarity of 100% with peptides of greater sequence already identified.

**Table 4 toxins-15-00264-t004:** Yield of generating anti-F2 fraction antibodies.

Immunization	Yield (U/mL)
1st immunization	0
2nd immunization	5733
3rd immunization	48,363
4th immunization	41,257
5th immunization	22,389
Final sangria	57,709

**Table 5 toxins-15-00264-t005:** Average rate of proteolytic inhibition reaction of BaV and F2 fraction.

Sample + Anti-F2 Fraction Ab + Substrate	Vm*	Statistical Difference
BaV + Ab + Abz-RPPGFRSPFR-QEDDnp	841	***
F2 Fraction + Ab + Abz-RPPGFRSPFR-QEDDnp	642	***
Control (BaV + Abz-RPPGFRSPFR-QEDDnp)	1823	ns
Control (F2 fraction + Abz-RPPGFRSPFR-QEDDnp)	7159	ns

In white-backed 96-well plates, two µg/well of the venom and the F2 fraction were applied with the anti-F2 fraction antibody and PBS pH 7.2. After 30 min of incubation at 37 °C, two µL/well was added of FRET Abz-RPPGFRSPFR-QEDDnp 250 µM substrate, in a final volume of 100 µL/well. Proteolytic activity was measured by spectrofluorimetry (λEX = 330 and λEM = 416 nm, reader: Hidex Sense, Hidex, Finland) for 15 min. The assay was performed in duplicate. Representative data from two independent trials. Representative data from two independent tests expressed as mean reaction rate (Vm* = mean reaction rate—concentration range/time interval) and analyzed by One-way ANOVA test followed by Dunnet’s Post Test. *p* < 0.001 (***). ns: not significant.

**Table 6 toxins-15-00264-t006:** mm^2^ of hemorrhage areas.

Sample	Venom (µg)	Hemorrhage (mm^2^)
A1	10 µg	231
A2	10 µg	263
B1	20 µg	287
B2	20 µg	579
C1	30 µg	319
C2	30 µg	340
D1	40 µg	650
D2	40 µg	396

Groups of mice (n = 20) were inoculated by intradermal injection with different concentrations of BaV. The diameter of the hemorrhagic area of each tissue fragment was plotted in the ImageJ 1.8.0 program and expressed in mm^2^.

**Table 7 toxins-15-00264-t007:** Percentage of Hemorrhagic Areas.

Sample	Venom (µg) + Anti-F2 Fraction Antibodies	Hemorrhage (mm^2^)	Medium % Inhibition	Statistical Difference
Control	10 µg venom in PBS	344 ± 103 mm^2^	-	-
Group A	10 µg venom + Ab 1:5	178 ± 73 mm^2^	48%	*
Group B	10 µg venom + Ab 1:10	75 ± 19 mm^2^	78%	**
Group C	10 µg venom + Ab 1:20	303 ± 107 mm^2^	13%	ns

Groups of mice (n = 16) were inoculated by intradermal injection with the DHM of 10 µg of BaV together with different concentrations of the anti-F2 fraction antibody, or PBS pH 7.2. The hemorrhagic halo was calculated in mm^2^ and mean percentage of inhibition, referring to the total diameter of neutralization by anti-F2 fraction antibodies. Representative data expressed as mm^2^ of bleeding area were analyzed by One-way ANOVA test followed by Dunnet’s Post Test. (*) Statistically different from the control. *p* < 0.05 (**). ns: not significant.

## Data Availability

The data obtained in this research are available for consultation through the email address kemily.godoi.esib@esib.butantan.gov.br. Data is not available on other sites for consultation.

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
