# Peer review of "Anti-Metalloproteases: Production and Characterization of Polyclonal IgG Anti-F2 Fraction Antibodies Purified from the Venom of the Snake Bitis arietans"

_toxins, 2023, doi:10.3390/toxins15040264_

Round 1

Reviewer 1 Report

The characterization of venoms and the evaluation of immunorecognition of antivenoms are important strategies to face snakebite envenomation.  Here, the authors present an interesting study focused on toxins from the venom of a medically important snake. The findings are relevant in the scientific context and with therapeutic implications. The manuscript must be carefully reviewed before publication.

1. Line 17: The results are not sufficient to determine that the purification of a toxin is fundamental for the development of antivenoms. Please limit the scope of the work to the presented findings.

2. Authors should reorganize the introduction avoiding one-sentence paragraphs.

3. The introduction is long with necessary description of results that are further explained.

4. Figure 1 and 2 can be presented together.

5. Quality of figure 1 should be improved.

6. What molecular weight marker is used in electrophoresis? Why are the masses approximate?

7. Snake toxins can be dimeric or oligomeric. It would be highly recommended to compare electrophoresis under reducing and non-reducing conditions.

8. How was protein abundance determined in the F2 fraction?

9. Authors should improve figure 3. I recommend using other tools for better quality.

10. Figure 3. Scientific name should be italicised.

11. Figure 5. O.D. or D.O.?

12. Table 9. How many times was the test performed? How many animals? What is the deviation?

13. The article has an excess of figures and tables, some showing similar results. In these cases, authors should organize some data as supplementary material.

14. The number of independent experiments and the statistical analysis must be highlighted in the figures and methodology.

 15. Why was the neutralization test not performed more realistically? Would it be recommended to evaluate the effectiveness of neutralization without prior incubation?

Reviewer 2 Report

The paper entitled Anti-metalloproteases: production and characterization of polyclonal IgG anti-F2 fraction antibodies purified from the venom of the snake Bitis arietans” contains potentially important data for the development and implementation of innovative diagnostic and therapeutic algorithms dedicated to acute poisoning toxicology or tropical medicine. Authors of this publication suggest – among others – that “…BaV contains toxins that trigger the hemorrhagic process, in addition to toxins that help in the spreading factor of the toxic activity of one of the main proteases of the venom, the metalloproteinases…”

Remarks:

1.     "Key Contribution" subsection does not answer – directly or even indirectly – the aim of the study – Therefore subsection "Conclusion" must be added.

2.     The resolution of the Figure 1., Figure 3., Figure 5C. is inadequate – resolution (of the mentioned Figures) must be improved.

3.     The rationale for the choice of the Bitis arietans venom should be provided.

4.     The article lacks of practical implications of the achieved results – must be added. 

5.     The article lacks of information on Bitis arietans venom standardization – it absolutely must be added.

6.     The article lacks of information on statistical methods – it absolutely must be added.

7.     According the last-mentioned remark – no information on statistical significance between the results demonstrated in "Figure 3." have been shown – must be added.

8.     Analogically, no information on statistical significance between the results demonstrated in "Figure 5C." have been shown – must be added.

Round 2

Reviewer 1 Report

The new version of this manuscript has been improved. Authors considered most of the reviewer's comments. Hence, I consider that this manuscript could be of interest for readers, particularly those looking for strategies to mitigate snakebite complications in rural settings, such as African countries. 

Reviewer 2 Report

Authors of the manuscript entitled "Anti-metalloproteases: production and characterization of polyclonal IgG anti-F2 fraction antibodies purified from the venom of the snake Bitis arietans" and submitted to the editor of the journal "Toxins", modified the manuscript.

Currently, the article is suitable for publication without corrections.